# Atypical Bacterial Pathogens and Small-Vessel Leukocytoclastic Vasculitis of the Skin in Children: Systematic Literature Review

**DOI:** 10.3390/pathogens10010031

**Published:** 2021-01-02

**Authors:** Céline Betti, Pietro Camozzi, Viola Gennaro, Mario G. Bianchetti, Martin Scoglio, Giacomo D. Simonetti, Gregorio P. Milani, Sebastiano A. G. Lava, Alessandra Ferrarini

**Affiliations:** 1Faculty of Biomedicine, Università della Svizzera Italiana, 6900 Lugano, Switzerland; celine.betti@eoc.ch (C.B.); vio.gennaro94@live.it (V.G.); mario.bianchetti@usi.ch (M.G.B.); martin.scoglio91@gmail.com (M.S.); giacomo.simonetti@eoc.ch (G.D.S.); alessandra.ferrarini@eoc.ch (A.F.); 2Department of Pediatrics, Pediatric Institute of Southern Switzerland, Ospedale San Giovanni, 6500 Bellinzona, Switzerland; milani.gregoriop@gmail.com; 3Department of Internal Medicine, Ente Ospedaliero Cantonale, 6500 Bellinzona, Switzerland; 4Pediatric Unit, Fondazione IRCCS Ca’ Granda Ospedale Maggiore Policlinico, 20100 Milan, Italy; 5Department of Clinical Sciences and Community Health, Università degli Studi di Milano, 20100 Milan, Italy; 6Pediatric Cardiology Unit, Department of Pediatrics, Centre Hospitalier Universitaire Vaudois and University of Lausanne, 1011 Lausanne, Switzerland; webmaster@sebastianolava.ch

**Keywords:** leukocytoclastic small-vessel vasculitis, atypical pathogens, *Mycoplasma pneumoniae*, *Legionella pneumophila*, *Chlamydophila pneumoniae*, *Chlamydophila psittaci*, *Coxiella burnetii*, *Francisella tularensis*

## Abstract

Leukocytoclastic small-vessel vasculitis of the skin (with or without systemic involvement) is often preceded by infections such as common cold, tonsillopharyngitis, or otitis media. Our purpose was to document pediatric (≤18 years) cases preceded by a symptomatic disease caused by an atypical bacterial pathogen. We performed a literature search following the Preferred Reporting of Systematic Reviews and Meta-Analyses guidelines. We retained 19 reports including 22 cases (13 females and 9 males, 1.0 to 17, median 6.3 years of age) associated with a *Mycoplasma pneumoniae* infection. We did not find any case linked to *Chlamydophila pneumoniae*, *Chlamydophila psittaci*, *Coxiella burnetii*, *Francisella tularensis*, or *Legionella pneumophila*. Patients with a systemic vasculitis (N = 14) and with a skin-limited (N = 8) vasculitis did not significantly differ with respect to gender and age. The time to recovery was ≤12 weeks in all patients with this information. In conclusion, a cutaneous small-vessel vasculitis with or without systemic involvement may occur in childhood after an infection caused by the atypical bacterial pathogen *Mycoplasma pneumoniae*. The clinical picture and the course of cases preceded by recognized triggers and by this atypical pathogen are indistinguishable.

## 1. Introduction

Respiratory infections caused by so-called atypical bacterial pathogens such as *Mycoplasma, Legionella*, and, less frequently, *Chlamydophila pneumoniae* or *psittaci*, *Coxiella burnetii* and *Francisella tularensis* are often accompanied by non-respiratory immunologically mediated features, which may involve almost all organ systems [1,2,3].

Non-granulomatous leukocytoclastic small-vessel vasculitis of the skin with immune deposits, subsequently referred to as cutaneous small-vessel vasculitis, are often preceded by an acute upper-respiratory tract infection and may be limited to the skin or involve other tissues [4,5]. Textbooks and reviews no more than marginally mention the association with a symptomatic respiratory disease caused by atypical bacterial pathogens. Since this issue has never been extensively covered, we systematically reviewed the literature. The purpose was to document the clinical features and the course in pediatric patients with cutaneous small-vessel vasculitis preceded by the mentioned pathogens.

## 2. Methods

### 2.1. Search Strategy

We performed a structured literature search [6] with no language or date limits in August 2020 on the Excerpta Medica, National Library of Medicine, and Web of Science databases following the Preferred Reporting of Systematic Reviews and Meta-Analyses guidelines [7]. Search terms were (“atypical pneumonia” OR “*Chlamydia pneumoniae*” OR “*Chlamydia psittaci*” OR “*Chlamydophila pneumoniae*” OR “*Chlamydophila psittaci*” OR “*Coxiella burnetii*” OR “*Francisella tularensis*” OR “*Legionella*” OR “*Mycoplasma pneumoniae*”) AND (“acute hemorrhagic edema” OR “anaphylactoid purpura” OR “Finkelstein-Seidlmayer disease” OR “Henoch purpura” OR “leukocytoclastic vasculitis” OR “immunoglobulin A vasculitis” OR “urticarial vasculitis” OR “small vessel vasculitis” OR “vasculitis”). The literature search was carried out by two investigators with the support of an experienced senior investigator, who independently screened titles and abstracts of all reports in a nonblinded fashion to remove irrelevant reports. Discrepancies in study identification were resolved by consensus. Subsequently, full-text publications were reviewed to decide whether the report fitted the eligibility criteria of the review. The bibliography of each identified report was also screened for secondary references.

### 2.2. Selection Criteria

Original articles published up to July 31, 2020, that reported on pediatric (≤18 years) cases of community-acquired upper or lower respiratory-tract infections caused by *Chlamydophila (Chlamydia) pneumoniae*, *Chlamydophila psittaci*, *Coxiella burnetii*, *Francisella tularensis*, *Legionella species*, or *Mycoplasma pneumoniae* temporally associated (by ≤2 weeks) with a non-granulomatous leukocytoclastic small-vessel cutaneous vasculitis were sorted. Only previously healthy subjects without any pre-existing chronic disease were included. Cases of vasculitis possibly resulting from an adverse drug reaction and cases with detectable circulating anti-cytoplasmic or anti-nuclear auto-antibodies were excluded.

The diagnosis of infection caused by an atypical pathogen was retained exclusively in cases with both a distinctive clinical presentation and an appropriate [1,2] microbiology laboratory testing.

Recognized criteria were used to confirm or infirm the vasculitis diagnosis made in the original reports. The diagnosis of systemic immunoglobulin A [4] vasculitis was retained in subjects presenting with palpable purpura and at least one of the following: abdominal involvement (pain, vomiting, high frequency or fluidity of bowel movements, intestinal bleeding, and intussusception), articular involvement (joint pain with or without swelling), or kidney involvement (pathological hematuria, with or without associated pathological proteinuria). The diagnosis of acute hemorrhagic edema vasculitis was made in well-appearing infants and young children with acute onset of erythematous annular skin lesions and pronounced diffuse nonpitting body edema [4]. A biopsy was not a prerequisite for the diagnosis of immunoglobulin A vasculitis or acute hemorrhagic edema vasculitis [4]. The diagnosis of urticarial skin-limited vasculitis was suspected in patients presenting exclusively with wheals that persist for >24 h, burn more than itch, and often leave residual purpura as they resolve [5]. On the other hand, the diagnosis of “unclassified” skin-limited cutaneous vasculitis was suspected in subjects presenting with palpable purpura, usually localized on the buttocks and legs without any other organ system involvement [8]. A skin biopsy disclosing a non-granulomatous neutrophil infiltration into small vessel walls with karyorrhexis was a prerequisite for the final diagnosis of urticarial skin-limited vasculitis.

### 2.3. Data Extraction—Case Assessment—Comprehensiveness of Reporting

A predesigned data extraction form was used to collect the following information of each included case into a worksheet: demographics; clinical data, laboratory features and biopsy studies; immunomodulatory drug management; and outcome. If needed, attempts were also made to contact authors of original reports to provide missing information. The **CAAR **score (for **C**utaneous, **A**bdominal, **A**rticular and **R**enal involvement) is widely recommended (Table 1) to measure the disease severity in immunoglobulin A vasculitis [4]. For the present review, we employed the CAAR score also in patients affected by a skin-limited disease. Comprehensiveness of reporting was evaluated using four items: 1. description of respiratory disease and microbiologic testing; 2. description of vasculitis features; 3. drug management; 4. time to recovery. Each item was rated as 0, 1, or 2 and the reporting comprehensiveness was graded according to the sum of each item as high (score ≥ 6), satisfactory (score 4–5), or low (score ≤ 3).

### 2.4. Data analysis

Categorical data are presented as frequency and were analyzed using the Fisher’s exact test. Continuous data are presented as median and interquartile range and were analyzed using the Mann–Whitney–Wilcoxon test [9,10]. Statistical significance was set at *P* < 0.05.

## 3. Results

### 3.1. Search Results

The systematic literature search returned 538 potentially relevant reports (Figure 1). After removing irrelevant reports, 76 full-text publications were reviewed for eligibility. For the final analysis, we retained 19 reports [11,12,13,14,15,16,17,18,19,20,21,22,23,24,25,26,27,28,29] including 22 cases, which were published after 1973 in English (N = 16) and Spanish (N = 3 ) from the following countries: Italy (N = 4), Japan (N = 3), Spain (N = 2), the United Kingdom (N = 2), France (N = 1), Belgium (N = 1), China (N = 1), Colombia (N = 1), the Czech Republic (N = 1), Greece (N = 1), Poland (N = 1), and South Korea (N = 1). All cases were temporally associated with a *Mycoplasma pneumoniae* infection. No cases were linked to *Chlamydophila pneumoniae*, *Chlamydophila psittaci*, *Coxiella burnetii*, *Francisella tularensis*, or *Legionella pneumophila*. Reporting comprehensiveness was high in 12, moderate in eight, and low in the remaining two cases.

### 3.2. Microbiologic Diagnosis

The laboratory diagnosis of *Mycoplasma pneumoniae* infection was made by detecting a significant rise in antibody titer when comparing acute and convalescent blood samples (N = 20), a positive *Mycoplasma* testing in a respiratory tract sample (N = 1), or both (N = 1).

### 3.3. Findings

In the 22 patients (13 females and 9 males, 1.0 to 17, median 6.3 years of age), the vasculitis syndrome was systemic and skin-limited in two-thirds and one-third of cases, respectively (Table 2). Patients with systemic vasculitis (8 females and 6 males, 6.1 (4.3–8.0) years of age) and patients with an isolated cutaneous vasculitis (5 females and 3 males; 5.0 (2.4–12.3) years of age) did not significantly differ with respect to gender and age. The cutaneous involvement was significantly (*P* < 0.05) less relevant in subjects with systemic vasculitis (mild, N = 12; moderate, N = 2; severe, N = 0) as compared with those affected by a skin-limited vasculitis (mild, N = 2; moderate, N = 2; severe, N = 4). Biopsy studies were performed in eight cases, as shown in Table 3.

The characteristics of the 14 patients with systemic immunoglobulin A vasculitis are given in Table 4. No kidney involvement was noted in eight, no abdominal involvement in five, and no articular involvement in three patients. None of the patients with kidney involvement was found to have an elevated creatinine level.

Oral (N = 2) or parenteral (N = 2) corticosteroids were prescribed in four cases. Dapsone was administered to a 4.0-year-old boy. The time to recovery was ≤12 weeks in all cases with this information.

Two of the eight patients affected by a skin-limited vasculitis were prescribed oral corticosteroids. The time to recovery was ≤4 weeks in seven and 6 weeks in one of these cases. This parameter was not statistically different between patients with systemic and skin-limited vasculitis.

## 4. Discussion

This careful literature review points out that, in childhood, a symptomatic community-acquired respiratory disease caused by an atypical bacterial pathogen may be followed by a skin-limited or, more frequently, a systemic immunoglobulin A small-vessel vasculitis. All cases follow a *Mycoplasma pneumoniae* infection, affect more females than males, and have a good prognosis.

Vasculitis was associated exclusively with a *Mycoplasma pneumoniae* infection. We do not have any clear-cut explanation for this observation. We tentatively offer two causes. First, immunologically mediated complications such as hemolytic anemia, urticaria or erythema multiforme are more commonly linked to *Mycoplasma pneumoniae* compared to the remaining atypical bacterial pathogens [3,4]. Second, atypical bacterial pathogens such as *Chlamydophila pneumoniae* or *psittaci*, *Coxiella burnetii* and *Francisella tularensis* are not a frequent cause of a community-acquired acute respiratory disease in childhood [3,4].

As in the present analysis, immunoglobulin A small vessel vasculitis generally is a benign disease with an excellent prognosis, which affects all ages but occurs more frequently in children between 3 and 11 years [4]. In the mentioned vasculitis, there is normally a slight gender predilection affecting males rather than females at a 2:1 ratio [4], at variance with the data observed in the cases preceded by an atypical bacterial pathogen that were included in this review.

The prevalence of small-vessel vasculitis of the skin accompanying a symptomatic respiratory infection caused by an atypical bacterial pathogen is presently unknown. Seven case series including a total of 1618 patients affected by an immunoglobulin A vasculitis never described this association [30,31,32,33,34,35,36]. On the other hand, a recent report from China [37] including 1200 pediatric cases of immunoglobulin A vasculitis found serological evidence for a recent group A Streptococcus infection in 205 (17%) and for a *Mycoplasma pneumoniae* infection in 58 (4.8%) cases. We do not have any exhaustive explanation for the discrepancy between the Chinese data [37] and the results of the aforementioned case series [30,31,32,33,34,35,36]. We speculate that pediatricians customarily do not test for atypical bacterial pathogens in patients with vasculitis, even if preceded by a lower respiratory-tract infection [4].

The vast majority of children affected by a skin-limited or systemic immunoglobulin A small vessel vasculitis recover spontaneously and may be cared for as outpatients. Management is predominantly supportive and includes adequate hydration and symptomatic relief of pain [4]. There is some evidence that glucocorticoids enhance the rate of resolution of arthritis and especially abdominal pain [4]. However, these measures do not appear to prevent recurrent disease [4,38]. Medical treatment of active kidney disease, which is rarely required and is discussed in detail elsewhere [38], depends upon whether the patient is a child or an adult and upon the severity of proteinuria and histologic lesions (especially the degree of crescent formation). Finally, there is a widespread view that antimicrobials speed up the recovery of atypical pneumonia but do not shorten the course of mucocutaneous complications [1,3].

Immunoglobulin A1 deposition is a crucial pathophysiologic feature in immunoglobulin A vasculitis [4]. In clinical practice, a skin biopsy is rarely performed, especially in childhood, because the diagnosis is normally made on a clinical basis. Immunoglobulin A deposits were not found in one case included in this review. This is likely related to the fact that deposits may disappear with time due to phagocytosis and proteolysis. Unsurprisingly, therefore, skin biopsy (if indicated) should be performed at the edge of fresh lesions to maximize the chance of finding immune deposits [39,40].

The results of this systematic review must be viewed with an understanding of the inherent limitations of the analysis, which included data from articles published over more than 40 years. Furthermore, a temporal association between an infection caused by *Mycoplasma pneumoniae* and the vasculitis onset does not inevitably imply a causal relationship. Finally, the analysis did not address atypical pneumonias precipitated by viral pathogens including, among others, adenoviruses, influenzaviruses, parainfluenzaviruses, respiratory syncytial virus, and especially coronaviruses. Available data suggest an association of severe acute respiratory syndrome coronavirus 2 with small-vessel vasculitis of the skin [41].

## 5. Conclusions

Respiratory diseases caused by atypical bacterial pathogens are notoriously associated with urticaria, erythema multiforme and, more rarely, erythema nodosum or varicella-like eruptions [1,3,42]. This systematic review of the literature indicates that also a cutaneous small-vessel vasculitis may occur after an infection caused by the atypical bacterial pathogen *Mycoplasma pneumoniae*. The clinical picture and the course are similar to those observed in cases preceded by infections such as common cold, tonsillopharyngitis, or otitis media [30,31,32,33,34,35,36].

## Figures and Tables

**Figure 1 pathogens-10-00031-f001:**
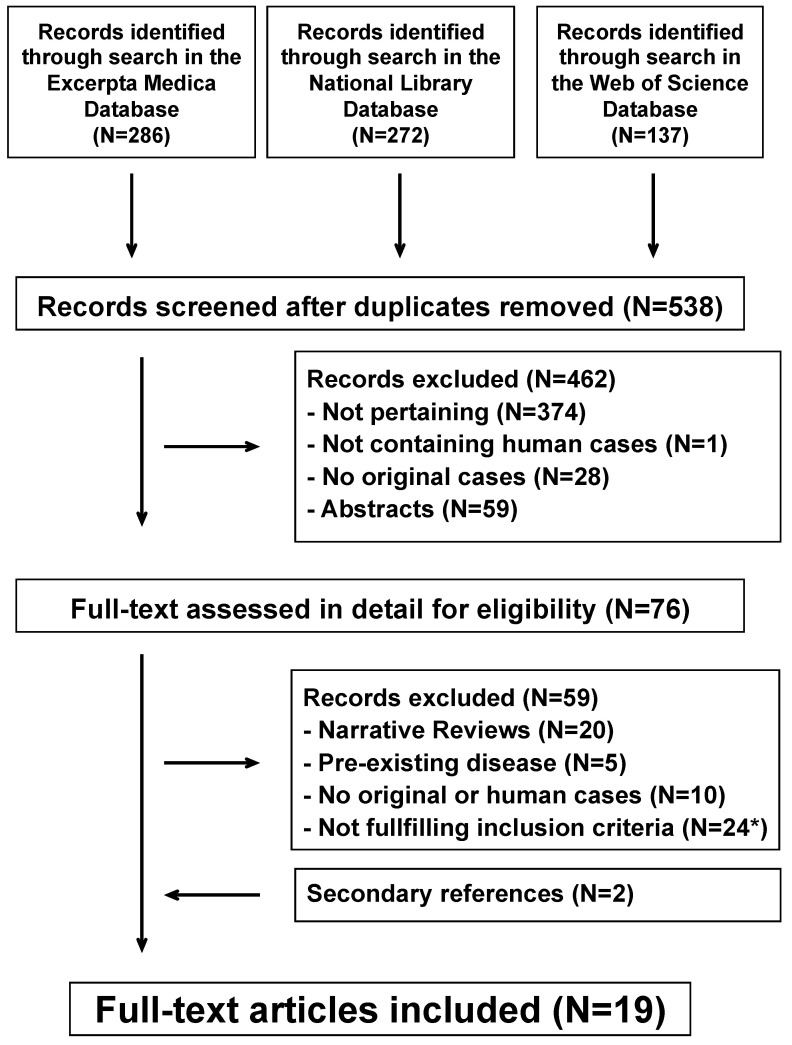
Small vessel leukocytoclastic vasculitis of the skin and atypical bacterial pathogens. Flowchart of the literature search process. *Including 10 adult cases (*Mycoplasma pneumoniae*, N = 7; *Chlamydophila pneumoniae*, N = 2; *Legionella pneumophila*, N = 1).

**Table 1 pathogens-10-00031-t001:** CAAR-grading system.

Organ’s involvement	
● Cutaneous involvement	
- absent: No eruption - mild: Eruptions located on buttocks and lower extremities alone - moderate: Eruptions located on (a) buttocks and lower extremities and (b) either trunk or upper extremities - severe: Eruptions located on (a) buttocks and lower extremities, (b) trunk and (c) upper extremities	
● Abdominal involvement	
- absent: No symptoms, no findings - mild: Mild abdominal pain (medically elicited) - moderate: Moderate abdominal pain (transient complaints brought to medical attention) - severe: Severe abdominal pain and/or melena, and/or hematemesis, and/or intussusception	
● Articular involvement	
- absent: No symptoms, no findings - mild: Symptoms or findings of articular involvement but no functional abnormalities - moderate: Symptoms and findings of articular involvement causing mild functional reduction (e.g., limping) - severe: Symptoms and findings causing moderate functional loss (e.g., inability to walk)	
● Renal involvement	
- absent: Normal urinalysis - mild: Pathological hematuria, normal proteinuria - moderate: Pathological hematuria, mild-moderate proteinuria - severe: Hematuria, severe proteinuria	

**Table 2 pathogens-10-00031-t002:** Characteristics of 22 patients, 1.0 to 17 years of age, affected by a cutaneous non-granulomatous leukocytoclastic small-vessel vasculitis with immune deposits preceded by a symptomatic respiratory disease caused by *Mycoplasma pneumoniae*.

Heading			All Cases
**N**			22
**Females:Males, N**			13:9
**Age, years**			
	years		6.3 [3.1–9.0]
**Respiratory disease**			
	Upper respiratory disease, N		13
	Lower respiratory disease, N		9
**Classification of vasculitis**			
	Systemic immunoglobulin A vasculitis, N		14
	Skin-limited vasculitis		
		Acute hemorrhagic edema vasculitis, N	3
		Urticarial vasculitis, N	1
		Unclassified, N	4
**Immunomodulatory** **drug treatment**			
	Systemic corticosteroids		6
	Dapsone		1

**Table 3 pathogens-10-00031-t003:** Results of testing for deposits of immunoglobulin A into skin-vessel walls on patients affected by a cutaneous non-granulomatous leukocytoclastic small-vessel vasculitis.

	Immunoglobulin AcairuiVasculitis	Skin-limited Vasculitis		
		Acute Hemorrhagic Edema	Urticarial	Unclassified
Skin biopsy performed, N	4 *	1	1	2
Testing for immunoglobulin A performed, N	3	1	1	1
Testing for immunoglobulin A positive, N	2	0	0	0

* Both a kidney and a skin biopsy were performed in 2 patients.

**Table 4 pathogens-10-00031-t004:** Characteristics of 14 pediatric patients (9 girls and 5 boys, aged 1.5 to 17, median 6.1 years of age) affected by a systemic immunoglobulin A vasculitis. The CAAR grading system was used.

Organ’s Involvement			N
			14
**Cutaneous involvement**			
	Mild		12
	Moderate		2
	Severe		0
**Abdominal involvement**			
	None		5
	Mild		3
	Moderate		4
	Severe		2
**Articular involvement**			
	None		3
	Mild		1
	Moderate		10
	Severe		0
**Kidney involvement**			
	None		8
	Mild		3
	Moderate		1
	Severe		2
**Immunomodulatory** **drug treatment**			
	Systemic corticosteroids		4
	Dapsone		1
**Time to recovery**			
	≤4 weeks, N		9
	5–12 weeks, N		4
	No information available, N		1

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
