# Peer review of "Atypical Bacterial Pathogens and Small-Vessel Leukocytoclastic Vasculitis of the Skin in Children: Systematic Literature Review"

_pathogens, 2021, doi:10.3390/pathogens10010031_

Round 1
Reviewer 1 Report
The authors describe the purpose of this study as documentation of clinical features and the course in patients with cutaneous small-vessel vasculitis preceded by a symptomatic respiratory disease caused by atypical bacterial pathogens.
The rationale behind the search strategy an hence the selection criteria is not clear. It should be explained and relevant references should be added. There is no estimation of expected coverage of the field or possible pitfalls with the strategy that has been chosen.
The CAAR-grading is only for assesment of organ involvement in IgA vasculitis.
The discussion does not really cover the field an the conclusion does not match the purpose ot the study.
Author Response
1) The rationale behind the search strategy and hence the selection criteria is not clear. It should be explained and relevant references should be added. There is no estimation of expected coverage of the field or possible pitfalls with the strategy that has been chosen.
We thank the Reviewer for this constructive comment. Some of us have a large experience with systematic literature reviews (and are authors of at least 20 corresponding reports). In the revised version of the manuscript, we provide some more information on the literature search process and provide a new reference (Krupski TL, Dahm P, Fesperman SF, Schardt CM. How to perform a literature search. J Urol. 2008 Apr;179(4):1264-70. doi: 10.1016/j.juro.2007.11.087). Furthermore, we explicitly state that the literature search was supervised by an experienced senior investigator.
2) The CAAR-grading is only for assessment of organ involvement in IgA vasculitis.
This comment is very interesting. The CAAR-grading system has been so far used uniquely for patients affected by IgA vasculitis. In the revised version of the manuscript, we explicitly state that we “exceptionally” employ this grading system also for patients affected by a skin-limited vasculitis.
3) The discussion does not really cover the field and the conclusion does not match the purpose of the study.
This comment is very constructive. The revised version of the section discussion and of the section conclusion is broader.
a) We discuss the discrepancy between the prevalence of immunoglobulin A vasculitis recently reported in China and the prevalence so far reported in case series.
b) The clinical picture and the course of vasculitides preceded by infections such as common cold, tonsillopharyngitis, or otitis media and vasculitis preceded by an atypical pathogen are discussed and compared.
Reviewer 2 Report
This article by C. Betti et al. describe an interesting review of the implication of atypical bacterial pathogens in leukocytoclastic vasculitis of the skin.
This review of little interest, except for dermatologists and include numerous modification needed before publication (some of them highlight unfinished work).
Title : is the title incomplete "and:"?
Methods
- Search strategy : how authors decided to share the work ? Is there a double verification ? What about discrepant decision?
- Authors have to give details about the secondary reference identification strategy.
- Line 63 : Have work done between June 31 and the date of literature search been excluded?
Results
- How was the diagnosis been done for direct identification of the microorganisms?
- What is the threshold retained for statistical association ? p<0.05 or p<0.03? Moreover, comparison between group, with a group including only 3 patient seem poorly appropriate (suppress in table 2 or add in table 4)
- No reference has to be included in the results part of the manuscript (line 147).
Conclusion :
- Line 199 : this manuscript describes a review and not a study.
Figures/Table :
- Improvement of the table polices and quality is needed. Tables are hardly readable for now.
- Figure 1 : More details have to be given of the study classified as : "not pertaining" as they represent three fourth of the screened records.
Author Response
1) Is the title incomplete “and:”?
Many thanks for this comment. In the revised manuscript, we removed the word “and”. The title of our report is: “Atypical bacterial pathogens and small-vessel leukocytoclastic vasculitis of the skin: systematic literature review”.
2) Search strategy: how authors decided to share the work? Is there a double verification? What about discrepant decision?
This comment is very pertinent. The initial version of the mentioned section was rather incomplete. In the revised version we provide information on the search strategy: a) unblinded by independent search by two of the authors; b) a senior author supervised the search; c) discrepancies solved by consensus.
3) Authors have to give details about the secondary reference identification strategy.
In the revised version of the manuscript we explicitly state: “The bibliography of each identified report was also screened for secondary references”.
4) Line 63: Have work done between June 31 and the date of literature search been excluded?
We thank the Reviewer for detecting this important typo. In the revised manuscript “June 31” was replaced by “July 31”.
5) How was the diagnosis been done for direct identification of the microorganisms?
This relevant information is provided in the section “3.2. microbiological studies” of the manuscript. Statement: “The laboratory diagnosis of Mycoplasma pneumoniae infection was made by … et cetera et cetera”. Please note that the diagnosis of infections caused by atypical pathogens (especially: Mycoplasma) is made serologically in the vast majority of cases. This attitude is well documented in the literature:
a) Meyer Sauteur PM, Unger WWJ, van Rossum AMC, Berger C. The Art and Science of Diagnosing Mycoplasma pneumoniae Infection. Pediatr Infect Dis J 2018;37:1192-1195.
b) Cunha BA. The atypical pneumonias: clinical diagnosis and importance. Clin Microbiol Infect 2006 ;12 (Suppl 3):12-24.
6) What is the threshold retained for statistical association? p<0.05 or p<0.03?
This comment is o.k. In the revised manuscript (section 2.4. Data analysis) we state: “Statistical significance was set at P<0.05”.
7) Comparison between group, with a group including only 3 patients seem poorly appropriate (suppress in table 2 or add in table 4).
This comment is interesting.
a) A group including only 3 cases is very small. However, the age is statistically different between the two groups because no “age overlap” was observed. Hence, we feel that the mentioned difference is clinically relevant.
b) However, we recognize that the information given in table 2 and table 4 is excessive. Hence, we reduced the number of columns in table 2 and table 4.
8) No reference has to be included in the results part of the manuscript (line 147).
We modified this section of the manuscript as suggested by this Reviewer.
9) Improvement of the table polices and quality is needed. Tables are hardly readable for now.
This comment is important. We simplified table 2 and table 4.
10) Figure 1. Not pertaining
This comment is constructive. We reworked Figure 1, and more explicitly specified exclusion criteria.
Round 2
Reviewer 1 Report
The manuscript has been improved and most comments from the previous review have been adressed.
Author Response
Dear Reviewer,
thanks for your precious help.
Best regards
Dr. Pietro Camozzi
Reviewer 2 Report
This article by C. Betti et al. describe an interesting review of the implication of atypical bacterial pathogens in leukocytoclastic vasculitis of the skin.
This review of little interest, except for dermatologists and include numerous modification needed before publication. Most of my previous comments have not been completely/correctly addressed, see thereafter.
5) How was the diagnosis been done for direct identification of the microorganisms?
This relevant information is provided in the section “3.2. microbiological studies” of the manuscript. Statement: “The laboratory diagnosis of Mycoplasma pneumoniae infection was made by … et cetera et cetera.” Please note that the diagnosis of infections caused by atypical pathogens (especially: Mycoplasma) is made serologically in the vast majority of cases. This attitude is well documented in the literature:
a) Meyer Sauteur PM, Unger WWJ, van Rossum AMC, Berger C. The Art and Science of Diagnosing Mycoplasma pneumoniae Infection. Pediatr Infect Dis J 2018;37:1192-1195.
b) Cunha BA. The atypical pneumonias: clinical diagnosis and importance. Clin Microbiol Infect 2006 ;12 (Suppl 3):12-24.
Even if this statement is important (and reference could be adedd to the manuscript.), my question focused on direct identification of microorganisms. Serology is an indirect detection of a infectious disease. Moreover, in both direct and indirect assays, authors have to discuss the heteregenity of the laboratory assays, as performances could be key for observations.
6) What is the threshold retained for statistical association? p<0.05 or p<0.03?
This comment is o.k. In the revised manuscript (section 2.4. Data analysis) we state: et cetera et cetera.”
This comment is important in the manuscript but the significance treshold in table 2 for example remained 0.03. Moreover, statistics could be of interest in table 4.
7) Comparison between group, with a group including only 3 patients seem poorly appropriate (suppress in table 2 or add in table 4.).
This comment is interesting.
a) A group including only 3 cases is very small. However, the age is statistically different between the two groups because no “age overlap” was observed. Hence, we feel that the mentioned difference is clinically relevant.
b) However, we recognize that the information given in table 2 and table 4 is excessive. Hence, we reduced the number of columns in table 2 and table 4.
à I maintained my previous comment. Three cases is too limited to highlight robust observations. It could not been eliminated that a sampling fluctuation is responsible for this observation. Please do not conclude on the "age" parameter, in the table and/or manuscript.
9)(suppress in table 2 or add in table 4.) Tables are hardly readable for now.
This comment is important. We simplified table 2 and table 4.
Moreover, please respect recommandation for police size and type for publication in pathogens (see table/figure recommandations)
10) Figure 1. Not pertaining
This comment is constructive. We reworked Figure 1, and more explicitly specified exclusion criteria.
More details have to be given about the "not pertaining" records, as they represent most of the excluded records.
Author Response
5. Serology is an indirect detection of an infectious disease. Moreover, in both direct and indirect assays, authors have to discuss the heterogeneity of the laboratory assays, as performances could be key for observations.
- This comment is very constructive. In the revised version of the manuscript, we shortly compare advantages and disadvantages of indirect and direct diagnosis of Mycoplasma pneumoniae infection. Furthermore, we acknowledge the broad heterogeneity of the techniques used.
6. What is the threshold retained for statistical association? p<0.05 or p<0.03?
- thanks. we fixed in p<0.05
7/9. Comparison between group, with a group including only 3 patients seem poorly appropriate (suppress in table 2 or add in table 4.). Tables are hardly readable for now
- Many thanks for these criticisms. We recognize that we modified the tables as suggested in your first review. Regrettably, we didn’t include the revised tables in the last submission.
10. More details have to be given about the "not pertaining" records, as they represent most of the excluded records.
- we preferred to keep the image unchanged to keep it easy to consult